# Emerging Antiarrhythmic Drugs for Atrial Fibrillation

**DOI:** 10.3390/ijms23084096

**Published:** 2022-04-07

**Authors:** Arnela Saljic, Jordi Heijman, Dobromir Dobrev

**Affiliations:** 1West German Heart and Vascular Center, Institute of Pharmacology, University Duisburg-Essen, 45147 Essen, Germany; arnela@sund.ku.dk (A.S.); jordi.heijman@maastrichtuniversity.nl (J.H.); 2Department of Biomedical Sciences, Faculty of Health and Medical Sciences, University of Copenhagen, 2400 Copenhagen, Denmark; 3Department of Cardiology, CARIM School for Cardiovascular Diseases, Maastricht University, 6229 ER Maastricht, The Netherlands; 4Department of Molecular Physiology & Biophysics, Baylor College of Medicine, Houston, TX 77030, USA; 5Montréal Heart Institute, University de Montréal, Montréal, QC H1T 1C8, Canada

**Keywords:** atrial fibrillation, pharmacology, ectopic activity, fibroblast

## Abstract

Atrial fibrillation (AF), the most common cardiac arrhythmia worldwide, is driven by complex mechanisms that differ between subgroups of patients. This complexity is apparent from the different forms in which AF presents itself (post-operative, paroxysmal and persistent), each with heterogeneous patterns and variable progression. Our current understanding of the mechanisms responsible for initiation, maintenance and progression of the different forms of AF has increased significantly in recent years. Nevertheless, antiarrhythmic drugs for the management of AF have not been developed based on the underlying arrhythmia mechanisms and none of the currently used drugs were specifically developed to target AF. With the increased knowledge on the mechanisms underlying different forms of AF, new opportunities for developing more effective and safer AF therapies are emerging. In this review, we provide an overview of potential novel antiarrhythmic approaches based on the underlying mechanisms of AF, focusing both on the development of novel antiarrhythmic agents and on the possibility of repurposing already marketed drugs. In addition, we discuss the opportunity of targeting some of the key players involved in the underlying AF mechanisms, such as ryanodine receptor type-2 (RyR2) channels and atrial-selective K^+^-currents (*I*_K2P_ and *I*_SK_) for antiarrhythmic therapy. In addition, we highlight the opportunities for targeting components of inflammatory signaling (e.g., the NLRP3-inflammasome) and upstream mechanisms targeting fibroblast function to prevent structural remodeling and progression of AF. Finally, we critically appraise emerging antiarrhythmic drug principles and future directions for antiarrhythmic drug development, as well as their potential for improving AF management.

## 1. Introduction

Atrial fibrillation (AF) is the most common cardiac arrhythmia worldwide [1]. Although AF is not an immediately life-threatening arrhythmia, it is associated with a number of clinically relevant complications such as an increased risk of stroke and heart failure, which contribute to the high morbidity and mortality rates amongst AF patients [2,3]. Despite significant advances, currently available therapies for the management of AF remain suboptimal. There is increasing evidence that rhythm control (i.e., maintaining normal sinus rhythm) may improve clinical outcomes [4,5,6], but current approaches have limited efficacy, partially due to a one-size-fits-most approach that ignores the diversity in the etiology and mechanisms underlying the different forms of AF [7,8]. Moreover, currently available pharmacological rhythm-control therapies are severely limited by potential proarrhythmic side effects [9]. Nevertheless, the increasing understanding of the complex cellular and molecular mechanisms of AF has revealed new potential targets and created opportunities for the development of novel pharmacological anti-AF therapies [10]. Traditionally, the primary focus in pharmacological AF therapy has been on the development of antiarrhythmic drugs (AADs) [11]. However, due to the increasing awareness about the progressive nature of the disease, the focus is now shifting to developing drugs targeting the AF-related atrial electrical and structural remodeling, which play an important role in the maintenance and progression of the arrhythmia. Here, we provide an overview of the key mechanisms involved in the pathophysiology of various forms of AF and discuss emerging developments of AADs that exploit our current understanding of AF mechanisms. Finally, we critically appraise principles and future directions for AAD development, as well as their potential for improving AF management.

## 2. Characteristics and Mechanisms of AF

### 2.1. Variability in AF Phenotypes

AF is considered a progressive disease. In most patients, the episodes are initially short and self-terminating, but if appropriate triggers and substrates are present, the episodes will progress and become more persistent and permanent [12]. AF is often associated with palpitations, reduced exercise tolerance and other symptoms, but symptom-rhythm correlation is variable [13,14]. Moreover, a significant number of patients are asymptomatic, despite being at increased risk for AF-associated adverse events [15]. Spontaneously converting AF lasting less than 7 days is classified as paroxysmal AF (pAF). AF that persists for more than 7 days is classified as persistent AF and if the arrhythmia persists for more than 12 months it is classified as long-standing persistent AF. When no further attempts are made to restore normal sinus rhythm, AF is classified as permanent [16]. In this review, long-standing persistent and permanent forms of AF are denoted as chronic AF (cAF). Finally, postoperative AF (POAF) is another common presentation of AF, occurring in up to 40% of cardiac and 20% of non-cardiac surgeries [17,18,19,20,21]. POAF most commonly occurs 2–4 days post-surgery and is usually characterized by short repetitive AF episodes [17,18,19,20,21].

### 2.2. General Mechanisms of AF

The initiation of an AF episode requires both a trigger and a substrate [12]. Focal spontaneous firing, also known as ectopic activity, represents the major AF-triggering mechanism [1,22,23]. Mechanisms leading to ectopic activity include enhanced automaticity, early afterdepolarizations (EADs) and delayed afterdepolarizations (DADs) [24,25,26]. In case of enhanced automaticity, the depolarizing current during diastole is large enough to depolarize the resting membrane potential across the threshold for action potential (AP) generation, causing ectopic firing [24]. Transient spontaneous depolarizations can also occur following complete AP repolarization, known as DADs. DADs may cause atrial ectopy if the depolarization is large enough to reach the threshold for Na^+^-channel activation and AP generation. The depolarizing current underlying DADs is created by increased Na^+^ influx through the Na^+^/Ca^2+^-exchanger (NCX), which is activated in response to spontaneous Ca^2+^-release events (SCaEs) from the intracellular stores of the sarcoplasmic reticulum (SR) through the ryanodine receptor type-2 (RyR2) channels [24,27]. Finally, if the AP duration (APD) is excessively prolonged, l-type Ca^2+^ channels can be reactivated, promoting a depolarizing current during the plateau phase or early in the repolarizing phase of the AP, creating EADs [24,28].

The evolution of the arrhythmogenic substrate due to atrial remodeling is promoted by cardiac and non-cardiac diseases, risk factors such as obesity and aging, and AF itself [12,29,30,31]. Atrial electrical remodeling includes Ca^2+^-handling abnormalities and changes in ion current properties that promote re-entry by abbreviating APD and effective refractory period (ERP) [32,33]. Atrial structural remodeling results from atrial fibrosis, atrial dilation and connexin changes that induce re-entry-promoting heterogeneous conduction slowing [34,35,36,37,38].

Depending on the type of AF, the key mechanisms for initiation and maintenance of the arrhythmia differ and the type and relative contribution of triggers as well as substrates change. The proper consideration of these different mechanisms is important for the discovery of specific targets and to enable tailored therapy for each type of AF. Figure 1 summarizes the leading mechanisms for each type of AF, which are discussed in greater detail in the following sections.

### 2.3. Fundamental Mechanism of pAF

Atrial cardiomyocytes from patients with pAF show DAD-mediated cellular triggered activity without clear evidence for electrical remodeling such as shortening of APD or hyperpolarization of the resting membrane potential [39]. The increased incidence of DADs in pAF patients is likely partly caused by an increased Ca^2+^ uptake into the SR by the SR Ca^2+^-ATPase type-2a (SERCA2a) leading to enhanced SR Ca^2+^ load, along with RyR2 dysregulation [40]. The open probability of RyR2 is higher in pAF [40], but in contrast to patients with cAF [41,42,43,44] this is associated with an increase in RyR2 expression but not channel hyperphosphorylation [40].

There is also evidence for the potential contribution of atrial fibrosis in pAF patients [35,45]. Compared to patients in sinus rhythm, pAF patients displayed atrial fibrosis in the area of crista terminalis [35,45] and the inferior pulmonary veins (PV) [35]. Similar trends were observed in the Bachman’s Bundle and left atrium, albeit no significant differences were found in these regions [35,45].

Ectopic activity arising from the PVs plays a particularly important role in pAF patients, with several potential proarrhythmic electrophysiological characteristics compared to the rest of the atria identified in animal models [46,47,48,49,50,51]. However, the molecular mechanisms promoting ectopic activity in the PVs of humans remain largely unknown due to the very limited access to tissue samples. Animal models may provide valuable insight into the molecular mechanisms of pAF [52], but relevant animal models with spontaneous development of paroxysmal AF are scarce and the PVs do not appear to play a major role for the induction and maintenance of AF in the majority of animal models [39]. Horses develop spontaneous AF, which can easily be detected by implantable loop recorders [53,54]. Spontaneous PV firing has been recorded in horses [55], suggesting similar pAF mechanisms in horses and humans. However, the cellular and molecular mechanisms of ectopy have not been assessed in horses so far. Nevertheless, horses could prove to be a valuable animal for studying the true nature of pAF.

Overall, the current evidence suggests that the leading arrhythmogenic mechanism in patients with pAF is DAD-mediated ectopic activity caused by Ca^2+^-handling abnormalities without clear evidence of atrial electrical remodeling. Although atrial fibrosis could contribute, the potential role of structural remodeling in pAF patients requires further investigation and validation.

### 2.4. Fundamental Mechanisms of cAF

The cellular and molecular determinants of re-entry and AF maintenance are best characterized in patients with persistent AF and in animal models with artificially maintained tachycardia to simulate the arrhythmia and to study the specific effects of the high atrial rate.

Electrical remodeling in cAF or due to atrial tachycardia causes rapid ERP/APD shortening [38], primarily by increasing K^+^ currents and reducing l-type Ca^2+^ current (*I*_Ca,L_) [56]. Although the agonist-dependent acetylcholine-activated inward-rectifier K^+^ current (*I*_K,ACh_) is decreased in cAF, it develops constitutive activity and, together with an increased basal inward-rectifier K^+^ current (*I*_K1_) and slow delayed-rectifier K^+^ current (*I*_Ks_), shortens APD and hyperpolarizes resting membrane potential, promoting re-entry [57,58,59,60,61,62]. The increase in *I*_K1_ is partly caused by increased protein expression of Kir2.1 [58,63] and enhanced single-channel open probability promoted by phosphatase-dependent channel dephosphorylation [64]. The constitutive activity of *I*_K,ACh_ in patients with cAF is due to increased phosphorylation by protein kinase C epsilon [64,65,66]. The functional expression of transient outward K^+^-current (*I*_to_) is lower in cAF patients, whereas *I*_Kr_ is unchanged [61,67,68,69].

Some studies also reported a reduced ultra-rapid delayed-rectifier K^+^ current (*I*_Kur_) in cAF patients [69,70], while other studies did not find current alterations [61,67,68,71]. Similarly, some studies revealed increased small-conductance Ca^2+^-activated K^+^ (SK) currents (*I*_SK_) in cAF, while other work showed a decreased *I*_SK_ [72,73,74,75]. Since the expression of SK channels and *I*_Kur_ is larger in atrial compared to ventricular tissue [72,76], and under physiological conditions SK channels play a minor role in the ventricles, these channels are considered atrial predominant anti-AF targets [77]. However, ventricular SK-channel expression is increased in heart failure and SK-channel inhibition can promote ventricular arrhythmogenesis under specific conditions [76]. The atrial-specific TASK-1 channel, a member of the two-pore domain K^+^ channel family, is also upregulated in cAF, contributing to APD shortening [78], but is downregulated in AF patients with concomitant left-ventricular dysfunction [79]. Thus, the degree of TASK-1 remodeling in AF patients is difficult to predict. Together, the alterations in these K^+^ currents likely contribute to the proarrhythmic shortening of the atrial ERP/APD in cAF patients.

The high atrial rates during AF promote a potentially toxic cellular Ca^2+^ overload. Consequently, *I*_Ca,L_ is reduced due to a calcineurin-dependent mechanism, contributing to re-entry-promoting APD shortening [80]. Increased nuclear Ca^2+^ releases via inositol 1,4,5-triphosphate with subsequent Ca^2+^-calmodulin protein kinase II (CaMKII) activation accelerates the nuclear export of histone deacetylase type-4 also contributing to the downregulation of *I*_Ca,L_ [81]. *I*_Ca,L_ downregulation in turn reduces the Ca^2+^-transient amplitude and atrial contractility [41]. Ca^2+^-handling abnormalities in cAF are primarily due to increased CaMKII activity, leading to RyR2 hyperphosphorylation at Ser2814 and leaky RyR2 channels causing potentially proarrhythmic SCaEs [41,42,82]. Protein kinase A (PKA)-mediated RyR2 hyperpolarization at Ser2808 has also been found in cAF patients [41,83], which promotes dissociation of the stabilizing calstabin2 (FKBP12.6) subunit [84,85]. SR Ca^2+^ load is unchanged in cAF patients despite the larger SR Ca^2+^ leak. This can either be due to enhanced phosphorylation of phospholamban [41,86] or a downregulation of the SERCA2a-inhibitory sarcolipin [87,88,89]. Increased PKA-mediated activation of the PP1-inhibitory protein inhibitor-1, which inhibits PP1 in the SR microdomain [90], could explain local phospholamban hyperphosphorylation [86].

Structural remodeling, particularly atrial fibrosis, makes important contributions to cAF and is largely irreversible [56]. Besides being caused by the underlying heart disease, atrial fibrosis could result from the high atrial rate during AF [91,92]. Together with connexin remodeling [38], collagen deposition due to atrial fibrosis creates re-entry-promoting conduction heterogeneities in cAF. Of note, myofibroblasts may also couple directly to cardiomyocytes through gap junctions, which may promote DAD-mediated triggered activity in cardiomyocytes.

### 2.5. Fundamental Mechanisms of POAF

Conceptually, the development of POAF requires a vulnerable preoperative substrate interacting with surgery-induced triggers such as activation of the autonomic nervous system [93] and inflammation [21]. The pre-existing substrate for POAF includes increased pro-inflammatory signaling, with the increased expression of activated components of the NLRP3 (NACHT, LRR and PYD domains-containing protein 3) inflammasome being the most prominent [26]. The NLRP3 inflammasome is a multiprotein complex responsible for the cleavage and release of the pro-inflammatory cytokines IL-1β and IL-18 [94,95]. Increased expression of NLRP3 inflammasome components was also identified in atrial cardiomyocytes from pAF and cAF patients, pointing to a common inflammatory signature in AF. In agreement, knock-in mice with cardiomyocyte-specific NLRP3 activation revealed an ERP abbreviation, likely due to an increase in *I*_Kur_, as well as atrial fibrosis and hypertrophy, both contributing to the increased AF susceptibility [96]. In addition, a higher frequency of SCaEs, likely due to CaMKII-dependent RyR2-hyperphosphorylation, along with *I*_Ca,L_ alternans were also noted in POAF patients Interestingly, DADs were absent under resting conditions and only became evident upon acute application of IL-1β, mimicking one of the major post-operative inflammatory triggers. Acute IL-1β application further promoted the CaMKII-dependent RyR2 hyperphosphorylation, thereby exacerbating the pre-existing Ca^2+^-handling abnormalities [26]. However, the exact consequences of enhanced NLRP3-inflammasome activity in other types of AF, as well as in patients with obesity and diabetes, who are at an increased risk of developing AF [97], remain incompletely understood.

Similar to patients with pAF, patients with POAF do not display changes in APD and have no major changes in K^+^ currents [26]. Thus, the leading mechanism in POAF appears to be ectopic (triggered) activity promoted by the acute inflammatory response without evidence of re-entry promoting APD and ERP changes, although structural changes such as connexin remodeling and fibrosis might also contribute, particularly in the left atrium [21,98,99,100].

## 3. Novel Pharmacological Approaches for AF Management

The current ESC guidelines on management of AF focus predominantly on appropriate anticoagulation, followed by symptom control with either rate or rhythm control and management of risk factors and comorbidities [16]. Initially, large clinical trials comparing rate and rhythm control such as Atrial Fibrillation Follow-up Investigation of Rhythm Management (AFFIRM) [101] and Rate Control vs. Electrical Cardioversion for Persistent Atrial Fibrillation (RACE) [102] did not find significant differences in outcome between rate and rhythm control. However, early modern rhythm control of AF might reduce the incidence of adverse cardiovascular outcomes, as shown in the Early Treatment of Atrial Fibrillation for Stroke Prevention Trial (EAST-AFNET 4) [5]. This has again highlighted the need for improved pharmacological options for rhythm control [103]. The challenge with traditional AADs is not only their modest efficacy [104], but rather their safety at the ventricular level, with ischemia and structural heart disease being a contraindication for most currently available AADs due to the high risk of drug-induced proarrhythmia. Of note, currently used drugs are not based on knowledge of the underlying arrhythmia mechanisms and none of these drugs have been developed to specifically target AF. With the growing understanding of the fundamental mechanisms of AF it will be likely possible to develop new and improved antiarrhythmic treatment modalities based on precise knowledge of the underlying mechanisms, including upstream therapy preventing electrical and structural remodeling.

### 3.1. Targeting Ectopic Activity

Cellular triggered (ectopic) activity is present in all forms of AF, although the silencing of atrial Ca^2+^-handling may also occur in cAF under specific conditions [105]. In vivo, atrial ectopy may play a more important role in pAF and POAF than for patients with cAF. As described in Section 2, RyR2 channels play a major role in the Ca^2+^-handling abnormalities typically seen in AF patients. One possibility for reducing ectopic activity is, therefore, targeting abnormal RyR2 channels either by using direct RyR2-channel blockers or RyR2-function modulators. Novel drugs in preclinical development are summarized in Table 1.

#### 3.1.1. RyR2-Channel Blockers

Besides being Na^+^-channel blockers, the Class Ic AADs such as flecainide and propafenone also inhibit RyR2 channels, reducing the open probability of RyR2 in vitro via an open-state blocking mechanism (i.e., shortened duration, but increased frequency of RyR2 channel openings, thereby increasing Ca^2+^ spark mass, while reducing the occurrence of spontaneous cell-wide Ca^2+^ waves), potentially preventing the formation of proarrhythmic Ca^2+^-dependent DADs [135,136] (Table 1). The antiarrhythmic actions of flecainide on RyR2 channels have also been demonstrated in a cardiac calsequestrin knockout catecholaminergic polymorphic ventricular tachycardia (CPVT) mouse model [106,137] (Table 1). Moreover, a flecainide analogue that lacks RyR2-blocking properties was unable to prevent CPVT [137], clearly indicating that the primary mode of action of flecainide in CPVT is inhibition of RyR2. In a small clinical trial including CPVT patients, flecainide, in combination with a β-blocker, was very effective in preventing exercise-induced ventricular ectopy [138]. In AF patients, novel routes of flecainide administration might increase cardioversion efficacy and compliance. In preclinical studies in pigs inhalative delivery of flecainide gives rise to a similar pharmacokinetic profile to flecainide given intravenously [139] and causes a rapid cardioversion of AF [107] (Table 2). The clinical potential of inhalative flecainide was recently tested in a clinical trial (Inhalation of Flecainide to Convert Recent Onset SympTomatic Atrial Fibrillation to siNus rhyThm (INSTANT), NCT03539302). The study included 101 patients with symptomatic recent onset AF (≤48 h). The highest dose of inhalative flecainide (120 mg) led to cardioversion in 48% of patients with a median time to conversion of 8.1 min after end of inhalation. Both the cardiac and extra cardiac adverse events were mild and transient and did not require treatment in any of the patients [140] (Table 2).

Ranolazine has been shown to reduce RyR2 open probability and to desensitize Ca^2+^-dependent RyR2 activation, thereby preventing Ca^2+^ overload and EADs [109] (Table 1). Ranolazine is used for treatment of angina pectoris and ischemic heart diseases and possess clear antiarrhythmic properties [149]. Unfortunately, ranolazine also carries an increased risk of proarrhythmia at the ventricular level by inhibiting *I*_Kr_ and prolonging APD and QT interval [149,150].

Carvedilol is a third-generation non-selective β-blocker with α_1_-receptor blocking properties which is used as antihypertensive drug, for the treatment of heart failure with reduced ejection fraction, and in patients with ventricular dysfunction following myocardial infarction [151,152,153,154]. Carvedilol and its analogues (VK-II-86, VK-II-36, CS-I-34, CS-I-59) also exhibit RyR2-blocking properties, reducing the incidence of both DADs and EADs [112,113] (Table 1). The effect of carvedilol may be partly due to its β-blocking and antioxidant actions, which may reduce RyR2 phosphorylation or oxidation [155,156]. Interestingly, the carvedilol analogues have no or minimal β-blocking properties, but still inhibit DAD-promoting SCaEs [112,113] due to a direct reduction in RyR2 open probability. Clinically used carvedilol consists of a mixture of S-and R-carvedilol enantiomers. In contrast to the S-enantiomer, R-carvedilol does not possess β-blocking properties, but specifically blocks RyR2 channels, preventing arrhythmias ex vivo and in vivo in mice [114]. Although the preclinical studies have shown positive signals (Table 1), there are no clinical data addressing the antiarrhythmic effects of carvedilol analogues (Table 2).

#### 3.1.2. RyR2-Channel Modulators

Besides direct RyR2-channel blockers, several RyR2 modulators are available or are under development. Dantrolene for instance is a RyR1 inhibitor and is clinically used to treat potentially life-threatening malignant hyperthermia [157,158]. Dantrolene has also been shown to stabilize cardiac RyR2 channels by improving the inter-domain interaction between the channel N-terminal and central domains [159,160] and by enhancing the binding between calmodulin and RyR2 [161], all of which exert antiarrhythmic effects [117,162,163]. In a sheep model of left atrial myocardial infarction dantrolene reduced the occurrence of spontaneous AF episodes [115] (Table 1). The effects of dantrolene have also been studied in isolated cardiomyocytes from patients with cAF, showing that dantrolene was able to reduce SR Ca^2+^ leak and suppress cellular DADs [116,117] (Table 1). Interestingly, when a CaMKII inhibitor (autocamtide-2-related inhibitory peptide, AIP) was applied in addition to dantrolene, no additive effects on the SR Ca^2+^ leak were observed. This either indicates that the effect on RyR2 is so strong that no additional benefit can be obtained by targeting CaMKII or that dantrolene has secondary direct or indirect effects on CaMKII activity. Dantrolene also reduced AF incidence in CaMKIIδ_C_ overexpressing mice [116]. In isolated cardiomyocytes from these mice dantrolene reduced SR Ca^2+^ leak and the incidence of both DADs and EADs [116]. These results call for additional studies investigating whether repurposing dantrolene could be an option for effectively treating AF.

Other RyR2 modulators are drugs that are capable of increasing the binding between RyR2 and its regulatory subunit calstabin2 (FKBP12.6), which acts like an endogenous RyR2-channel inhibitor by stabilizing the closed state of the channel [84,85,164]. S107 inhibited the dissociation of calstabin2 from RyR2 [165,166], reduced SR Ca^2+^ leak in mouse atrial cardiomyocytes and lowered the incidence of inducible AF in vivo in mice [118] (Table 1). Although these drugs have been in preclinical development for more than a decade, there are still no clinical data available (Table 2).

#### 3.1.3. CaMKII Inhibitors

Drugs targeting CaMKII are of potential interest because they target multiple effector proteins including RyR2 [167]. There are drugs in preclinical development with promising efficacy for atrial arrhythmias such rimacalib (SMP-114), RA608, hesperadine which are available for oral administration. RA608 is an ATP-competitive CaMKII inhibitor selectivity targeting the predominant cardiac forms, namely CaMKIIδ and CaMKIIγ [168]. The ATP-competitiveness of RA608 is a great advantage, as it enables the drug to bind both the active and inactive forms of CaMKII. Application of RA608 to human atrial cardiomyocytes reduces CaMKII-dependent SR Ca^2+^ leak and prevents induction of AF in mice [120]. Rimacalib is also an ATP-competitive CaMKII inhibitor, which is currently evaluated in a clinical trial for the treatment of rheumatoid arthritis. Rimacalib reduced Ca^2+^ SR leak in human atrial and ventricular cardiomyocytes [169,170,171] (Table 1). Hesperadin is an ATP-competitive selective CaMKIIδ-inhibitor that was recently shown to improve ischemia/reperfusion induced cardiac damage [119]. However, at present, no clinical trials are ongoing to support the efficacy of these drugs in AF patients (Table 2).

### 3.2. Targeting ERP/APD Changes

Class III AADs such as amiodarone, dofetilide and sotalol are K^+^ channel blockers that prolong ERP, thereby destabilizing re-entry. However, many Class III AADs (e.g., vernakalant, amiodarone, and its derivates dronedarone, celicarone, budiodarone and SAR11464A) are multi-channel blockers with effects from all AAD classes [104]. As mentioned above, the safety profiles of these AADs at the ventricular level is a major limitation hindering the development of novel AADs. The challenge with targeting the typical ERP/APD changes in cAF patients is to design drugs that prolong atrial ERP without clinically relevant pro-arrhythmic effects in the ventricles, especially in patients with ischemia and structural heart diseases, which are very common in AF patients. For this purpose, many atrial-selective targets such as *I*_SK_, *I*_K2P_, *I*_Kur_ and *I*_K,ACh_ are still being evaluated with the hope of establishing an atrial-selective anti-AF option with reasonable safety at the ventricular level and outside the heart.

#### 3.2.1. *I*_SK_ Channel Blockers

The antiarrhythmic properties of SK channels have been investigated in several animal models utilizing a number of different compounds [52,172]. However, the expression of the three SK-channel subtypes differs greatly between species. Moreover, SK channels undergo complex remodeling in response to a wide range of pathophysiological stimuli [173] and the reduction in SK current during cAF observed in some studies may limit the efficacy of SK-channel blockers in certain subpopulations of cAF patients [72]. Thus, careful evaluation of SK-channel inhibitors in well-defined patient populations will be critical. The currently leading SK-channel blocker candidate, AP30663, has proven to be safe for use in humans in phase-1 studies and has entered phase-2 clinical trials [141] (Table 2). In a pig model of tachypacing-induced AF, the cardioversion rate of AP30663 in vernakalant-resistant animals was found to be 60%. AP30663 prolonged aERP in a dose-dependent manner. The pigs also displayed a dose-independent decrease in heart rate as wells as a dose-dependent prolongation of QTc, which could be attributed to inhibition of *I*_Kr_ [121] (Table 1). In the phase-1 clinical trial AP30663 did not affect QRS duration, but prolonged QTc by 18.8 ± 4.3 ms, which was considered a mild and transient effect [141] (Table 2).

Besides the selective SK channel blockers, other drugs such as ondansetron and enalapril also have SK-channel inhibiting properties [174]. SK channels blockers have also been used in combination with other AADs such as amiodarone, dofetilide, flecainide and ranolazine and have shown promising results in animal models. Although the HARMONY trial has previously shown a positive outcome with combined AAD therapy using dronedarone and ranolazine [175], this approach still needs extensive clinical investigation and further validation.

#### 3.2.2. TASK-1 Channel Blockers

One of the recently added atrial selective targets for potential rhythm control therapy is TASK-1. Two already well-known anti-arrhythmic drugs, namely, vernakalant and amiodarone, both inhibit TASK-1 channels [176]. Selective TASK-1 inhibitors have also been developed and have been used in preclinical investigations. Recent in vivo and ex vivo investigations in pigs, cellular recordings from isolated human atrial cardiomyocytes, and computational modeling suggests that in the atria, the respiratory stimulant doxapram [177,178] acts as a selective TASK-1 blocker with strong antiarrhythmic properties [123] (Table 1). Doxapram is currently being investigated in the DOxapram Conversion To Sinus rhythm (DOCTOS) trial for cardioversion of both paroxysmal and persistent non-valvular AF (EudraCT No: 2018-002979-18) (Table 2).

#### 3.2.3. *I*_Kur_ and *I*_K,ACh_ Channel Blockers

Although preclinical investigations suggested very promising anti-arrhythmic properties for a range of *I*_Kur_ and *I*_K,ACh_ inhibitors, their clinical efficacy has been rather disappointing [179] and, therefore, most of the clinical development programs have been stopped [180]. The lack of efficacy for *I*_Kur_ blockers can be partly attributed to a reduction in *I*_Kur_ in some cAF populations [70,181]. In contrast to *I*_Kur_, *I*_K,ACh_ has been shown to be constitutively active during cAF [65]. Although *I*_K,ACh_ blockers have good antiarrhythmic efficacy in several large AF animal models including dog [182,183,184], goat [185] and horse [186], the conducted clinical trials did not demonstrate a clear antiarrhythmic efficacy [184,187]. The GIRK channels underlying *I_K_*_,ACh_ are not expressed in the ventricles and are one of the main mediators of parasympathetic regulation, which could promote both the initiation and maintenance of AF [24]. Some *I*_K,ACh_ drug-development programmes were stopped because of drug-related extra-cardiac toxicity which precluded the testing of the drug’s efficacy for AF. Thus, inhibition of *I*_K,ACh_ still has the potential to become a clinically effective anti-AF option and some anti-*I*_K,ACh_ drugs are still under development.

### 3.3. Targeting Upstream Mechanisms

#### 3.3.1. Targeting Atrial Profibrotic Signaling

Atrial fibrosis is an important determinant of re-entry, especially in patients with cAF. One crucial step in the increased production of fibrosis is the proliferation and differentiation of fibroblasts into collagen-secreting profibrotic myofibroblasts [92]. In general, two approaches are possible for targeting fibroblast differentiation and related fibrogenesis. One possibility is to directly slow down the differentiation of fibroblast into active myofibroblasts. An alternative approach is to target upstream pathways, such as inflammatory signaling, which promotes fibroblast differentiation.

Differentiation of fibroblast into myofibroblasts is mediated by a number of cytokines and growth factors [188]. One approach for slowing fibroblast differentiation is to target some of these mediators. A central cytokine that is involved in the activation of fibroblasts in atria is the transforming growth factor-β1 (TGF-β1) [189,190,191,192,193,194]. TGF-β1 activates the Smad transcription factors and induces expression of fibroblast differentiation-specific genes [195]. In a transgenic mouse model, overexpression of TGF-β1 increased atrial fibrosis but not ventricular fibrosis, resulting in slowed and heterogeneous atrial conduction and increased AF inducibility [196]. Pirfenidone is a drug used to treat idiopathic pulmonary fibrosis with both anti-inflammatory and anti-fibrotic properties, in part by reducing the expression of TGF-β1 [197]. TGF-β1 was upregulated in a canine model of congestive heart failure (3 weeks of ventricular tachypacing) leading to atrial fibrosis and increased AF vulnerability. These effects could be prevented by pirfenidone administration during the 3 weeks of tachypacing [124] (Table 1). Similarly, in vivo inhibition of TGF-β1 with neutralizing antibodies reduced fibrosis and prevented cardiac dysfunction in pressure-overloaded rats [126,127]. The TGF-β1 inhibitor GW788388 is currently in preclinical development [125] (Table 1).

Another approach for inhibiting fibroblast differentiation is to target fibroblast ion channel function. Fibroblasts express transient receptor potential canonical-3 (TRPC3) channels, which regulate cellular Ca^2+^ entry and are activated upon stretch, thereby controlling fibroblast function [188,198]. Harada et al. investigated the role and expression of TRPC3 in AF patients, AF goats and dogs with tachypacing-induced heart failure. In all models, the expression of TRPC3 was upregulated, promoting fibroblast proliferation and differentiation. In the canine model, pyrazole-3, a TRPC3 inhibitor, prolonged the ERP, decreased fibroblast proliferation and the expression of extracellular matrix-related genes, along with a reduction in AF duration [128] (Table 1). TRP melastatin-related 7 (TRPM7) channels are similarly upregulated in fibroblasts isolated from AF patients compared to sinus rhythm controls. Knockdown of TRPM7 in cultured fibroblasts reduced the TRPM7-mediated Ca^2+^ influx and inhibited the TGF-β1-induced differentiation. Although the TRPM7 current increases in patients with AF the expression does not seem to be altered, pointing to altered channel function rather than expression in cAF [199]. Recently, LBQ657, a metabolite of sacubitril, was shown to block the TRPM7 current and prevent fibroblast activation in vitro [200]. Sacubitril is an angiotensin receptor neprilysin inhibitor used in combination with valsartan in heart failure patients [201] (Table 1). Currently, drugs targeting fibroblast ion channel function are only available for pre-clinical research and none are in clinical development (Table 2).

Polo-like kinase 2 (PLK2) in a serine-threonine kinase regulating cell proliferation, mitochondrial respiration and apoptosis [202,203,204]. Expression of PLK2 is reduced in AF patients. PLK2 knockout mice exhibit increased interstitial fibrosis and AF susceptibility. Both in human atrial samples and in the mouse model, a reduction in PLK2 results in ERK1/2 phosphorylation and secretion of the cytokine-like glycoprotein osteopontin to the extracellular matrix [130]. Mesalazine, an anti-inflammatory drug used to treat inflammatory bowel disease [205], also inhibits ERK1/2 and could potentially be repurposed for use in the cardiovascular field (Table 1).

Another important kinase for fibroblast function is AMP-activated kinase (AMPK), which provides cross-talk between different fibrotic signaling pathways. For instance, activated AMPK inhibits the TGF-β1/Smad signaling pathway and prevents the differentiation of fibroblast into myofibroblasts [206]. Reduced AMPK activity plays a central role in the development of renal fibrosis [207] and has also been associated with diabetes mellitus, obesity and aging [208,209,210], all of which also predispose to AF. Metformin, an antidiabetic drug, is a strong activator of AMPK [211]. The anti-fibrotic effects of metformin were demonstrated in a mouse model of bleomycin-induced lung fibrosis and attributed to deactivating differentiated myofibroblasts by AMPK-dependent activation of autophagy [212]. Whether similar deactivation mechanisms are possible for cardiac fibroblasts is unknown. Observational studies indicate that metformin is associated with a lower risk of AF compared to other oral antidiabetic drugs such as sulfonylureas [213]. Several clinical trials are currently investigating the antiarrhythmic properties of metformin (Table 2), but no preclinical studies are available to demonstrate the antiarrhythmic efficacy of metformin (Table 1).

Cross-linking of collagen and elastin is mediated by lysyl oxidase (LOX) and LOX-like isoforms, which are believed to play an important role in the remodeling of the extracellular matrix [214]. Protein expression of LOX and LOX-like isoforms are found in a range of cardiac diseases and cross-linking of collagens increases myocardial stiffness [215] thereby contributing to cardiac remodeling. cAF patients have a higher expression of cross-linked collagen and LOX compared to both patients in sinus rhythm and patients with pAF [131,174]. Thus, although LOX inhibitors are expected to exert antifibrotic effects, further experimental and clinical work is needed to test this hypothesis (Table 1).

#### 3.3.2. Targeting Atrial Inflammatory Signaling

Targeting inflammatory signaling, which is upstream of atrial fibrosis, has emerged as a therapeutic option for patients with POAF, in which the acute postoperative inflammatory response also acts as a trigger of atrial arrhythmogenesis. However, NLRP3 inflammasome activity is also increased in pAF and cAF patients and could contribute to substrate progression over time. Thus, targeting inflammatory signaling could potentially be beneficial in all forms of AF. However, extensive preclinical and clinical studies are required to demonstrate and validate the inflammatory hypothesis of AF (Table 1 and Table 2).

Targeting of key cytokines such as NLRP3-inflammasome derived IL-1β and IL-18 as well as TNFα is an emerging option for AF management. IL-1β and TNFα inhibitors such as anakinra and etanercept are currently under clinical investigation, but for indications other than cardiac diseases [94]. Canakinumab, a monoclonal IL-1β antibody, reduced major cardiac events in patients with atherosclerosis in the CANTOS trial (The Canakinumab Anti-Inflammatory Thrombosis Outcome Study) [216]. Recently, the pilot clinical trial CONVERT-AF investigated the recurrence rate of AF after electrical cardioversion of patients with persistent AF in the absence or presence of canakinumab [142]. Although the conventional level of statistical significance was not reached (*p* = 0.09), AF recurrence at 6 months occurred in 77% and 36% of patients in the placebo and canakinumab groups, respectively [142]. The promising results of this pilot trial require further proof and verification in future studies with either canakinumab or the many highly selective small molecules targeting the assembly and activation of the NLRP3-inflammasome components that are currently in preclinical development [94].

Another anti-inflammatory drug that has been used for decades is colchicine. Colchicine is a microtubule-disrupting drug used for prophylactic treatment of gout and familial Mediterranean fever [217]. The assembly of the NLRP3 inflammasome is driven by the spatial arrangement of the tubule network and by disrupting this network colchicine is able to inhibit the assembly and activation of the NLRP3-inflammasome complex [94,218]. Long-term use of low-dose colchicine has been shown to be safe and the use of colchicine within the cardiovascular filed has gained great interest over the last two decades [219]. Colchicine has been tested for prevention of POAF following both open-heart surgery and ablation in several small clinical trials (Table 2). Some of these trials have shown reduced incidence of AF [146,220], while others obtained neutral results [144,145] (Table 2). However, the study design differed greatly between these studies in terms of drug loading dose, onset of treatment (pre/post-surgery), and observation time. Therefore, additional trials are warranted to demonstrate the potential value of colchicine in the different forms of AF.

## 4. Challenges and Future Perspectives of Antiarrhythmic Drug Development

The development of novel AADs faces many challenges and there are many obstacles for the translation from basic discoveries to clinical applications [103]. Besides the high regulatory hurdles and the industry concerns, the strategies used in contemporary AAD development also need careful consideration. First, the complexity of the clinical condition in which AF develops requires a tailored approach already by first clinical exposure of a novel AAD in order to have a real chance to show a positive signal. Safety concerns also play a major role because almost all available and novel AADs possess a proarrhythmic potential. Therefore, atrial selectivity combined with minimal non-cardiac toxicity is often considered the most promising approach for the development of novel AADs for AF. However, many of the putative anti-AF targets presented above are not exclusively present in the atria. For example, although inhibition of CaMKII could be a promising anti-AF approach, CaMKII is expressed in many tissues and its inhibition could give rise to serious extra-cardiac side effects. Thus, successful drug design and appropriate formulation and application need to be carefully considered in early stage of drug development to ensure high atrial selectivity and minimal extra-cardiac effects, particularly in the brain. Atrial-specific gene therapy would be the ultimate option to ensure appropriate selectivity and besides concerns and issues that need consideration in the clinical setting an atrial-specific adeno-associated serotype 9 (AAV9) vector has already been successfully developed and used in mice to generate efficient and atrial-specific gene expression following a single dose of systemic delivery [221]. The development of RNA-based therapeutics is also quickly emerging.

The most promising atrial-selective drugs are those targeting TASK-1 and SK channels. TASK-1 channel inhibitors have shown promising results in preclinical studies [123] and the first clinical trial using this approach is ongoing. Although SK channels are highly expressed in the central nervous system [222], the small molecules currently under development do not seem to pass the blood–brain barrier and do not exert serious neurological adverse effects in animal models [121] or in the Phase I clinical trial [141]. However, SK channels are not exclusively expressed in the atria of humans [76], suggesting only moderate atrial selectivity of SK channel inhibition. Perhaps because of this a QT prolongation was noted in the Phase I study [141], making further clinical investigations necessary to demonstrate the efficacy and validate the safety of SK-channel blockers.

Currently, no selective RyR2 blockers are available. Because RyR2 receptors are highly expressed in the central nervous system and are particularly important for learning and memory [223], drugs with clear RyR2-blocking properties such as flecainide could be of value if their delivery to the atria could be optimized to reduce ventricular and extra-cardiac side effects (Table 2). Although, in theory, CaMKII inhibition could have strong antiarrhythmic efficacy, its large abundance in the brain and other organs might preclude its use for AF. For instance, CaMKII plays a key role in neural plasticity and memory [224]. Furthermore, the γ isoform of CaMKII is crucial for female fertility. Female CaMKIIγ knock-out mice show infertility due to defects in egg activation [225]. CaMKIIδ and γ isoforms are both present in the heart, with the δ isoform being approximately 2.5-fold more prevalent [226]. Potential novel CaMKII inhibitors should therefore ideally be CaMKIIδ-isoform selective and should not pass the blood–brain barrier to reduce serious extra-cardiac side effects.

Targeting fibrosis for cardiac indications has not translated into clinical application. This is especially true for AF, although the atria are particularly prone to structural remodeling. The slow regression of fibrosis also argues for early intervention to prevent structural remodeling, well before AF is detected. In addition, inhibition or reversion of fibrosis might impair ventricular integrity and could cause ventricular dysfunction. Furthermore, the extracellular matrix is the main scaffold of many other organs such as liver, kidney and lungs, which might be adversely affected by a systematically applied anti-fibrotic drug. Again, atrial selectivity might be crucial for antifibrotic drugs and future studies dissecting the organ differences in fibroblast function and activation are clearly warranted. Upstream therapy, specifically targeting the NLRP3 inflammasome or cytokines, might constitute a novel therapeutic option. The CONVERT-AF trial showed a non-significant trend for canakinumab in reducing the recurrence rate of AF in patients with persistent AF that underwent electrical cardioversion [142] (Table 2), which supports the notion that anti-inflammatory approaches might be effective against AF. Further clinical studies testing the anti-inflammatory hypothesis of AF are required in order to develop novel AADs with anti-inflammatory properties.

## 5. Conclusions

Despite significant advances in catheter ablation therapy, AADs will remain a cornerstone of rhythm control therapy for millions of AF patients. Recent studies have significantly improved our understanding of the molecular mechanisms underlying different forms of AF, identifying novel opportunities for developing mechanism-based antiarrhythmic approaches including AF-specific ion-channel blockers, targeting of Ca^2+^-handling abnormalities (CaMKII, RyR2), and modulation of upstream pathways (notably inflammatory signaling). Numerous challenges, particularly related to achieving atrial selectivity and identifying the optimal (combination of) targets for different forms of AF, still need to be overcome and well-designed clinical trials are needed to confirm safety and efficacy of these novel approaches. Nevertheless, the rapid developments in preclinical studies on AF therapies suggest that there is still hope for novel pharmacological approaches for the most common clinically relevant arrhythmia.

## Figures and Tables

**Figure 1 ijms-23-04096-f001:**
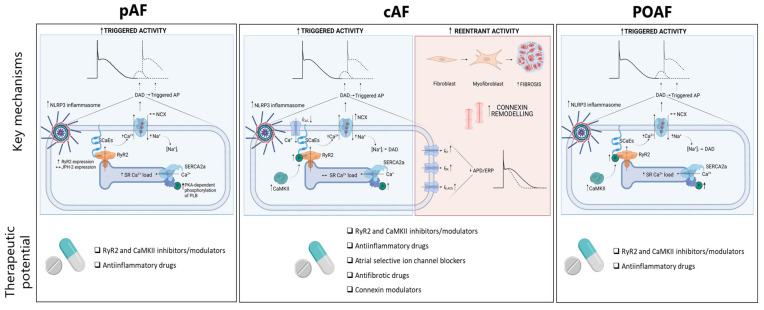
Key pathways for formation of AF. Triggered activity is common in all 3 forms of AF, which includes Ca^2+^ handling abnormalities, RyR2-channel dysfunction and triggered activity in form of DADs. Re-entry activity is commonly observed in patients with cAF, but less frequent and consistent in patients with POAF and pAF. The NLRP3 inflammasome is upregulated in all forms of AF, but plays the biggest role as an acute trigger in patients that present with POAF. AP, action potential; APD, action potential duration; cAF, chronic atrial fibrillation; CaMKII, Ca^2+^-calmodulin protein kinase II; DAD, delayed afterdepolarization EAD, early afterdepolarization; JPH-2, junctophilin-2; PKA, protein kinase A; ERP, effective refractory period; NCX, Na^+^/Ca^2+^-exchanger; NLRP3 inflammasome, NACHT, LRR and PYD domains-containing protein 3; pAF, paroxysmal AF; PLB, phospholamban; POAF, post-operative AF; RyR2, ryanodine receptor type-2; SCaE, spontaneous Ca^2+^-release events; SERCA, sarcoplasmic reticulum Ca^2+^-ATPase type-2a; SR, sarcoplasmic reticulum.

**Table 1 ijms-23-04096-t001:** Experimental evidence of antiarrhythmic efficacy of emerging drugs for atrial fibrillation.

Compound	Target	Mode of Action	Study	Animal	Antiarrhythmic Effects
**Flecainide**	RyR2 channels	Reduces open probability of RyR2 channels by an open-state blocking mechanism that increases Ca^2+^ spark mass, but reduces frequency of RyR2-mediated cell-wide Ca^2+^ waves. The antiarrhythmic properties of flecainide can also largely be attributed to its the Na_v_1.5 blocking effect	Watanabe et al. (2009) [106]	Mouse	Flecainide prevents arrhythmias in a mouse model of CPVT by inhibiting RyR2-mediated Ca^2+^-release events
		↓DADs	Verrier et al. (2018) [107]	Pig	Inhalable flecainide causes rapid (3.5–6.5 min) AF cardioversion
**R-propafenone**	RyR2 channels	Reduces open probability of RyR2 channels by an open-state blocking mechanism that increases Ca^2+^ spark mass, but reduces frequency of RyR2-mediated cell-wide Ca^2+^ waves. The antiarrhythmic properties of flecainide can also largely be attributed to its the Na_v_1.5 blocking effect	Faggioni et al. (2014) [108]	Mouse	R-propafenone prevents AF induction in calsequestrin 2 knockout mice
		↓DADs		In vitro	R-propafenone reduces frequency, amplitude and propagation speed of Ca^2+^ waves in isolated atrial myocytes from calsequestrin 2 knockout mice. In the same cells, R-propafenone reduces the incidence of pacing induced spontaneous Ca^2+^ waves and prevents triggered beats
**Ranolazine**	RyR2 channels	Reduces open probability of RyR2 channels, desensitizes Ca^2+^-dependent RyR2 activation and prevents cytosolic Ca^2+^ oscillations. The antiarrhythmic properties of flecainide can also largely be attributed to its the Na_v_1.5 blocking effect	Parikh et al. (2012) [109]	Rabbit	Ranolazine suppresses EADs in Langendorff-perfused rabbit hearts
		↓EADs	Carstensen et al. (2018) [110] Carstensen et al. (2019) [111]	Horse	Ranolazine displays a 25% cardioversion rate in horses with acutely induced AF and does not change the atrial ERP. Ranolazine reduces the atrial fibrillatory just before cardiversion
**Carvedilol, ** **R-carvedilol, ** **Carvedilol analogues: ** **VK-II-86,** **VK-II-36,** **CS-I-34** **and CS-I-59**	RyR2 channels	Dual effect caused by β-AR block and antioxidant actions that reduce phosphorylation and oxidation of RyR2, along with an open state channel block, all decreasing RyR2-mediated Ca^2+^-release events	Zhou et al. (2011) [112]	Mouse	Analogues prevent stress-induced VT in RyR2-mutated mice
			Maruyama et al. (2014) [113]	Mouse Rabbit	VK-II-36 inhibits VTs by preventing EADs and DADs in Langendorff-perfused hearts
		↓EADs and DADs	Zhang et al. (2015) [114]	Mouse	R-carvedilol suppresses spontaneous Ca^2+^ waves and CPVT in RyR2-mutated mice
**Dantrolene**	RyR2 channels	Stabilizes the close-state of RyR2 channels by improving the interaction between the *N*-terminal and central domains and by enhancing the binding between calmodulin and RyR2	Avula et al. (2018) [115]	Sheep	Dantrolene suppresses spontaneous AF episode in AMI sheeps
			Pabel et al. (2020) [116]	Mouse	Dantrolene suppresses AF inducibility in mice overexpressing CaMKIIδC
		↓EADs and DADs	Hartmann et al. (2017) [117]	In vitro	Dantrolene reduces SR Ca^2+^ spark frequency and diastolic SR Ca^2+^ leak in human atrial AF and ventricular HF cardiomyocytes, but does not affect the APD in these cardiomyocytes
**Rycal S107**	RyR2 channels	Stabilizes the interaction between calstabin2 (FKBP12.6) and RyR2, reducing open probability of RyR2	Shan et al. (2012) [118]	[119] Mouse	S107 reduces diastolic SR Ca^2+^ leak and decreases AF inducibility
		↓DADs			
**Rimacalib** **(SMP-114)**	CaMKII	ATP-competitive CaMKII inhibitor	No preclinical studies focusing on rimacalib and atrial arrhythmias		
**Hesperadin**	CaMKIIδ	ATP-competitive CaMKIIδ-inhibitor.	Zhang et al. (2022) [119]	Mouse	Hesperadin improves I/R- and overexpressed CaMKII-δ9-induced myocardial damage and HF in mice and stem cell-derived cardiomyocytes
**RA608**	CaMKII	ATP-competitive CaMKII inhibitor. Reduces SR Ca^2+^ leak, diastolic tension and increased SR Ca^2+^ content	Mustroph et al. (2020) [120]	Mouse	Oral RA608 significantly reduces inducibility of atrial and ventricular arrhythmias in CaMKIIδ transgenic mice 4 h after administration
				In vitro	RA608 reduced SR Ca^2+^ leak, diastolic tension and increased SR Ca^2+^ content
**AP30663**	SK channels	SK channel inhibitor	Diness et al. (2020) [121]	Pig	AP30663 cardioverts vernakalant-resistant AF and prevents AF reinduction
		↑ERP			
		↓Substrate for re-entry	Bentzen et al. (2020) [122]	Guinea pig Rat	AP30663 prolongs atrial ERP in isolated guinea pig hearts and anaesthetized rats
**Doxapram**	TASK-1 channels	TASK-1 channel inhibitor	Wiedmann et al. (2021) [123]	Pig	Doxapram cardioverts induced AF and prevents occurrence of inducible AF. Doxapram given daily prevents AF induced shortening of atrial ERP
		↑APD and ↑ERP			
		↓Substrate for re-entry			
**Pirfenidone**	TGF-β1	TGF-β1 inhibitor reducing fibroblast activation, atrial fibrosis and structural remodeling	Lee et al. (2006) [124]	Canine	Pirfenidone attenuates the arrhythmogentic left atrium remodeling by reducing conduction heterogeneity and atrial fibrosis, thereby decreasing duration of inducible AF
		↓Substrate for re-entry			
**GW788388**	TGF-β receptor-1	Selective TGF-β receptor-1/ALK5 inhibitor. Reduction in structural remodeling.	Oliveira et al. (2012) [125] Ferreira et al. (2019) [126]	Mouse	GW788388 reverses the loss of Cx43 and reduces cardiac fibrosis in models of Chagas’ heart disease
		↓Substrate for re-entry	Tan et al. (2009) [127]	Rat	GW788388 attenuates systolic dysfunction, phosphorylated Smad2, and reduces α-SMA and collagen I in a rat model of MI
**Pyrazole-3**	TRPC3	TRPC3 inhibitor preventing fibroblast proliferation and atrial fibrosis	Harada et al. (2012) [128]	Canine	Pyrazole-3 suppresses AF, increases ERP, and reduces fibroblast proliferation and atial fibrosis
		↓Substrate for re-entry			
**LBQ657 ** **(a sacubitril metabolite)**	TRMP7	TRPM7 inhibitor preventing fibroblast proliferation and atrial fibrosis	Li et al. (2020) [129]	Rabbit	LBQ657 reverses atrial enlargement, fibrosis, atrial ERP shortening and decreasesAF inducibility. It decreases collagen I and III, NT-proBNP, ST2, calcineurin, and prevents the downregulation of Ca_v_1.2
		↓Substrate for re-entry			
**Mesalazine**	ERK	Inhibitor of ERK-phosphorylation	Künzel et al. (2021) [130]	Mouse	Mesalazine normalizes OPN expression and prevents atrial fibrosis and dilation PLK2 knockout mice
		↓Substrate for re-entry			
**Metformin**	AMPK	Activates AMPK and inhibits differentiation of fibroblasts, thereby reducing cardiac fibrosis	No preclinical studies focusing on metformin and atrial arrhythmias		
		↓Substrate for re-entry			
**Statins**	Rac1/LOX	HMG-CoA reductase inhibitor. Inhibits Rac1 activation	Adam et al. (2011) [131]	Mouse	Statin reduces LOX expression, deposition of insoluable collagen and collagen cross-linking
		↓Substrate for re-entry			
**MCC950**	NLRP3 inflammasome	Inhibits ASC oligomerization and NLRP3 assembly	Yao et al. (2018) [96]	Mouse	MCC950 attenuates spontaneous premature atrial contractions and incidence of inducible AF in knock-in mice with cardiomyocyte-restricted NLRP3 activation
		↓Atrial ectopy and ↓Substrate for re-entry			
**Colchicine**	NLRP3 inflammasome	Microtubuli disruption preventing NLRP3 inflammasome assembly	Wu et al. (2020) [132]	Rat	Colchicine reduces the duration of inducible AF and prevents AF reinduction in rats with sterile pericarditis, along with a reduction in neutrophil infiltration, expression of IL-6, TGF-β, and TNF-α, atrial fibrosis and fibrosis-related genes, and signaling molecules (STAT3, P38, and AKT)
		↓Substrate for re-entry			
**Canakinumab**	IL-1β	Neutralizing monoclonal IL-1β antibody	No preclinical studies focusing on canakinumab and atrial arrhythmias		
		↓Substrate for re-entry (?)			
**Rilonacept**	IL-1 receptor	IL-1 decoy receptor	No preclinical studies focusing on rilonacept and atrial arrhythmias		
		↓Substrate for re-entry (?)			
**Anakinra**	IL-1 receptor	IL-1R antagonist	De Jesus et al. (2017) [133]	Mouse	Anakinra improves conduction velocity, decreases APD and APD dispersion, Ca^2+^ alternans and prevents pacing induced ventricular arrhythmia in a MI model. Anakinra also preserves the Cx43 expression and increased SERCA expression in the model
		↓Substrate for re-entry (?)			
**Etanercept**	TNFα	TNFα decoy receptor inhibitor.	Aschar-Sobbi et al. (2015) [134]	Mice	Etanercept abolishes exercise-induced NFκB-driven increases in gene transcription and reduces atrial fibrosis and the susceptibility to AF
		↓Substrate for re-entry			

AF, atrial fibrillation; AMI, acute myocardial infarct; AMPK, AMP-activated protein kinase; APD, action potential duration; ATP, adenosine triphosphate; CaMKII, Ca^2+^-calmodulin protein kinase II; CPVT, catecholaminergic polymorphic ventricular tachycardia; Cx43, connexin 43; DAD, delayed afterdepolarization; EAD, early afterdepolarization; ERK, extracellular signal-regulated kinase; ERP, effective refractory period; HF, heart failure; I/R, ischemia/reperfusion; IL-6, interleukin 1; IL-6, interleukin 6; IL-1β, interleukin 1β; LOX, lysyl oxidase; NLRP3 inflammasome, NACHT, LRR and PYD domains-containing protein 3; NT-proBNP, N-terminal-proB-type natriuretic peptide; OPN, osteopontin; PLK2, polo like kinase 2; RyR2, ryanodine receptor type-2; SERCA, sarcoplasmic reticulum Ca^2+^-ATPase type-2a; SR, sarcoplasmic reticulum; SERCA, sarcoplasmic reticulum Ca^2+^-ATPase type-2a; SR, sarcoplasmic reticulum; αSMA, α smooth muscle actin; SR, sarcoplasmic reticulum; TGF-β1, transforming growth factor-β1; TNFα, tumor necrosis factor α; TRPC3, transient receptor potential canonical-3; TRPM7, transient receptor potential melastatin-related 7; VT, ventricular fibrillation.

**Table 2 ijms-23-04096-t002:** Clinical evidence of antiarrhythmic efficacy of emerging drugs for atrial fibrillation.

Compound	Study	Aim	Patient Cohort	Outcomes	Adverse Events	Potential Use in AF
**Flecainide** **acetate**	NCT03539302 INSTANT Crijens et al. (2022) [140]	Inhalation of flecainide for cardioversion of recent onset symptomatic AF	101 patients with symptomatic AF	48% cardioversion rate within 90 min	Cardiac adverse events were uncommon including post-conversion pause, bradycardia and AFL. Extra-cardiac adverse events were mild and transient and included cough, throat, pain, throat irritation	POAF, pAF
**Flecainide** **acetate**	NCT05039359 RESTORE-1	Inhalation of flecainide for cardioversion of recent onset symptomatic AF	Recruiting	Phase 2 ongoing	See above	POAF, pAF
**R-propafenone**	NCT02710669	Comparison of R- and S- propafenone for prevention of AF recurrence following AF ablation procedure	Terminated (study halted/terminated prematurely due to COVID)	Terminated	Terminated	POAF, pAF
**AP30663** **(SK channel blocker)**	NCT04571385 Gal et al. (2020) [141]	Evaluating the efficacy and safety of AP30663 for AF cardioversion	47 healthy male volunteers	Phase 1 completed Phase 2 ongoing	Phase 1: Concentration dependent increase in the QTc interval (+18.8 ± 4.3 ms for highest dose)	POAF, pAF, cAF
**Doxapram (TASK-1** **channel blocker)**	2018-002979-17 DOCTOS	Use of doxapram as a new antiarrhythmic drug for atrial-selective AF therapy	Recruiting	Phase 1 and 2 ongoing	Trial ongoing	POAF, pAF, cAF
**Canakinumab**	NCT01805960 CONVERT-AF Krisai et al. (2020) [142]	Use for canakinumab for the prevention of recurrent AF after electrical cardioversion in patients with persistent AF	24 patients (11 placebo and 13 receiving canakinumab)	Canakinumab caused a trend of non-significant reduction in AF recurrence rate	One infection-related hospitalization	pAF, cAF
**Metformin**	NCT04625946	Metformin for prevention of recurrent atrial arrhythmias after ablation	Recruiting	Phase 4 ongoing	Trial ongoing	pAF, cAF
	NCT03603912 TRIM-AF	Metformin and/or lifestyle/risk factor modifications to reduce AF burden and AF progression	Recruiting	Phase 4 ongoing	Trial ongoing	pAF, cAF
	NCT02931253	Metformin as an upstream therapy for AF prevention after catheter ablation	Terminated (recruitment issues: enrollment expectations not met)	Terminated	Terminated	pAF, cAF
**Colchicine**	Deftereos et al. (2012) [143]	Colchicine for prevention of AF recurrence following PVI	161 (80 placebo and 81 receiving colchicine)	Colchicine reduces recurrence of AF at 3-month follow-up (16% occurrence in colchicine group vs. 33.5% in placebo group)	Gastrointestinal adverse effects	POAF, pAF
	Zarpelon et al. (2016) [144]	Colchicine for prevention of POAF in patients undergoing myocardial revascularization	140 (71 placebo and 69 receiving colchicine)	Colchicine does not reduce the incidence of POAF (7% occurrence in colchicine group vs. 13% in placebo group *p* = ns).	Postoperative infection (26.8% in the colchicine group vs. 8.7% in the placebo group *p* = 0.007)	POAF, pAF
	NCT03021343 END-AF Tabbalat et al. (2016) [145]	Colchicine for prevention of AF in patients undergoing open heart surgery	360 patients (181 placebo and 179 receiving colchicine)	Colchicine does not reduce the incidence of POAF (14% occurrence in colchicine group vs. 20% in placebo group *p* = ns).	Gastrointestinal adverse effects	POAF, pAF
	COPPS-POAF Imazio et al. (2011) [146]	Colchicine for prevention of POAF	360 (180 placebo and 180 receiving colchicine)	Colchicine reduced POAF (12% occurrence in colchicine group vs. 22% in placebo group *p* = 0.021)	Gastrointestinal adverse effects (*p* = 0.082)	POAF, pAF
	NCT01552187 COPPS-2 Imazio et al. (2014) [147]	Colchicine for prevention of post-pericardiotomy syndrome and POAF	360 patients (180 placebo and 180 receiving colchicine)	Colchicine reduced post-pericardiotomy syndrome, but not the development of POAF (33.9% occurrence in colchicine group vs. 41.7% in placebo group *p* = ns)	Gastrointestinal adverse effects	POAF, pAF
	NCT01985425 COP-AF Pilot Bessissow et al. (2018) [148]	Colchicine for prevention of perioperative AF in patients undergoing open heart surgery	100 patients (51 placebo and 49 receiving colchicine)	New AF/flutter occurred in 5 (10.2%) patients in the colchicine group and 7 (13.7%) patients in the placebo group (*p* = 0.76)	Few (nausea and vomiting) and similar in both groups	POAF, pAF
	NCT03310125 COP-AF	Colchicine for prevention of perioperative AF in patients undergoing open heart surgery	Recruiting	Phase 3 ongoing	Trial ongoing	POAF, pAF
	NCT04224545 COCS	Colchicine for prevention of AF after cardiac surgery in the early post-operative phase	Recruiting	Phase 4 ongoing	Trial ongoing	POAF, pAF
	NCT04155879	Colchicine for prevention of AF recurrence following electrical or pharmacological cardioversion	Not yet recruiting	Not yet recruiting	Not yet recruiting	POAF, pAF
	NCT04160117 IMPROVE-PVI Pilot	Colchicine for improvement of patient related outcomes after AF catheter ablation	Recruiting	Phase 2 ongoing	Trial ongoing	POAF, pAF
	NCT04906720 PAPERS	Colchicine for reduction in post-AF ablation induced pericarditis and reduction in POAF	Recruiting	Phase 2 ongoing	Trial ongoing	POAF, pAF
	NCT04870424 Co-STAR	Colchicine for prevention of new-onset AF after transcatheter aortic valve implantation	Recruiting	Phase 3 ongoing	Trial ongoing	POAF, pAF
	NCT02260206	Colchicine for prevention of AF recurrence after acute pericardial effusion following catheter ablation	Status unknown	Status unknown	Status unknown	POAF, pAF
	NCT02177266	Colchicine for prevention of post-pericardiotomy syndrome and AF	Terminated (difficulties in patient recruitment)	Terminated	Terminated	POAF, pAF
	NCT03015831 END-AFLD	Colchicine for prevention of POAF following cardiac surgery	Terminated (statistical analysis of interim data showed no benefit of colchicine)	Terminated	Terminated	POAF, pAF
	NCT02582190 COPPER-AF	Colchicine for prevention of AF recurrence post electrical cardioversion	Withdrawn (investigational medicinal product supplies)	Withdrawn	Withdrawn	POAF, pAF

cAF, chronic atrial fibrillation; pAF, paroxysmal atrial fibrillation; POAF, post-operative atrial fibrillation; AFL, atrial flutter; PVI, pulmonary vein isolation.

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
