# Peer review of "Emerging Antiarrhythmic Drugs for Atrial Fibrillation"

_ijms, 2022, doi:10.3390/ijms23084096_

Round 1

Reviewer 1 Report

Saljic et al. made a reasonably comprehensive review on emerging therapies in AF. I have some minor comments that might further improve this review:

1. Mechanisms underlying AF are important for the topic, yet they are not it's main focus and could therefore be somewhat shorter.

2. In order to increase readability of the manuscript, the authors should provide us with a comprehensive table including all the novel therapies used (mechanism, human data, pitfalls...)

3. Conclusion section should be expanded with future perspectives addressing which of all these presented medications (why and how) could most likely be seen in clinical practice in the future years.

Reviewer 2 Report

This is well-written review manuscript. However my main concern is the title of the manuscript. I do not think that the title "Emerging pharmacological approaches to treat atrial fibrillation" is representative of the manuscript. I thing that this manuscript mainly describes the drug antiarrhythmic in atrial fibrillation. Emerging pharmacological approaches for AF should also contain anticoagulation and rhythm control therapy. Furthermore the word "treat"is misleading. Is there any therapy to treat AF? I do not think so...AF is a chronic disease. Consequently, the authors should decide which is the purpose of this review and change/form the title of the manuscript appropriately.

Round 2

Reviewer 2 Report

I think that the authors have adequately addressed our comments.